# Characterization of Traditional Croatian Household-Produced Dry-Fermented Sausages

**DOI:** 10.3390/foods9080990

**Published:** 2020-07-24

**Authors:** Tina Lešić, Nada Vahčić, Ivica Kos, Manuela Zadravec, Blanka Sinčić Pulić, Tanja Bogdanović, Sandra Petričević, Eddy Listeš, Mario Škrivanko, Jelka Pleadin

**Affiliations:** 1Croatian Veterinary Institute, Laboratory for Analytical Chemistry, Savska Cesta 143, 10000 Zagreb, Croatia; lesic@veinst.hr; 2Faculty of Food Technology and Biotechnology, University of Zagreb, Pierottijeva 6, 10000 Zagreb, Croatia; nvahcic@pbf.hr; 3Faculty of Agriculture, University of Zagreb, Department of Animal Science and Technology, Svetošimunska Cesta 25, 10000 Zagreb, Croatia; ikos@agr.hr; 4Croatian Veterinary Institute, Laboratory for Feed Microbiology, Savska Cesta 143, 10000 Zagreb, Croatia; zadravec@veinst.hr; 5Administrative Department of Agriculture, Forestry, Hunting, Fishery and Water Management, Šetalište Pazinske Gimnazije 1, 52000 Pazin, Croatia; blanka.sincicpulic@istra-istria.hr; 6Croatian Veterinary Institute, Regional Veterinary Institute Split, Poljička Cesta 33, 21000 Split, Croatia; t.bogdanovic.vzs@veinst.hr (T.B.); petricevic.vzs@veinst.hr (S.P.); e.listes.vzs@veinst.hr (E.L.); 7Croatian Veterinary Institute, Regional Veterinary Institute Vinkovci, Ul. Josipa Kozarca 24, 32100 Vinkovci, Croatia; skrivanko@veinst.hr

**Keywords:** meat products, traditional sausages, nutritional quality, sensory evaluation, lipids, fatty acids, superficial micobiota, molds

## Abstract

Characterization of five types of traditional Croatian dry-fermented sausages produced by family farms was performed via identification of superficial mycobiota, physicochemical, sensory, instrumental color, fatty acids & fat quality indices. Detailed characterization of these sausages aimed to achieve standardization of their production and composition and to establish and/or improve their specification protocols. Traditional sausages varied significantly (*p* < 0.05) in all analyzed parameters except for the number of mold isolates. Sausages coming from eastern Croatia had a greater mold species diversity, with the highest number of isolated mycotoxigenic species in Slavonian domestic sausage. Sensory evaluation showed good acceptability of all sausages. According to health recommendations, Kulenova Seka showed the most representable values for most of fat quality indices. The results suggest the need for certain modifications in fat & fatty acid composition and, to a lesser extent, in salt content, however not at the expense of product safety, quality and acceptability.

## 1. Introduction

Naturally fermented sausages have a long production tradition in many European countries [1,2,3]. They are considered to be the products of top sensorial quality and represent a significant source of proteins, while meat quality combined with traditional production technology provides for their unique sensory properties [4,5]. In Croatia, there are many different types of natural dry-fermented sausages, poorly investigated insofar and widely varying in their qualities depending on the ingredients used in their formulation and the processing conditions. Non-standardized production of dry sausages in small-scale facilities also contributes to their heterogeneous quality. Non-industrial sausage production is driven by cultural practices related to the production region, non-regulated and variable production conditions, seasonality of the production and its small scale [5,6].

The surface of dry-fermented sausages is prone to mold colonization. Identification of molds during production is necessary, as some of them have an important production role and beneficial effects, such as the development of specific flavor and taste due to lipolytic and proteolytic activities, antioxidation through catalase activity, oxygen consumption and light shielding assuring color stabilization and delayed rancidity, protection against pathogenic or spoiling microorganisms, water loss reduction and sausage skin peeling easing. On the other hand, some mold species can be responsible for undesirable effects, such as food spoilage or compromising through antibiotic and mycotoxin contamination, responsible for consumer allergic reactions, antibiotic resistance and acute & chronic toxicity [7,8,9,10].

Characterization of similar traditional products from other countries such as Italy, Spain, Greece, Portugal, Slovakia, Belgium and France—in terms of their nutritional and sensorial properties—has been a subject of a number of studies [1,2,3,4,11,12,13]. Apart from aiming to achieve better characterization of final products, these studies also strived to contribute to the definition of unique quality markers and the improvement of product specification protocols, essential when dealing with products of the protected designation of origin (PDO) or protected geographic indication (PGI) [14]. Studies of house mycobiota of traditional dry-cured meat products contribute to the complete characterization of these products [15,16,17,18]. Their determination in regional products is very important for producers as an indicator of sausage quality that can be recognized and associated with the region of origin sausage identity [18]. In Croatia, only two studies identifying surface molds overgrowing dry-cured meat products were performed [9,19].

Health-related nutrition aspects of meat products have also been emphasized. Healthy trends in meat-product consumption give preference to low-fat and low-salt products [14,20]. Consumption of animal fats is related to the excessive intake of saturated fatty acids and the increased ratio of n-6 over n-3 polyunsaturated fatty acids (n-6/n-3 PUFA) associated with the development of cardiovascular and other chronic diseases [20]. Dry-cured products have an excessive salt content facilitating the development of hypertension, cardiovascular diseases and stomach cancer, so that recommendations were made to reduce salt consumption [5,21]. These changes must be achieved without any expense of sensory properties, quality and safety of the products in reference.

The aim of this study was the characterization of yet unexplored, but also the most common types of Croatian traditional dry-fermented sausages produced on households without use of starter cultures via identification of superficial mycobiota and determination of their physiochemical composition, instrumental color, sensory and fatty acid profile and fat-quality indices. The research results will contribute to the standardization of the composition and processing conditions of the products under study. The study also aims to identify the possibilities of product quality improvement, so as to preserve and maintain consumer acceptance developed based on favorable health impact and desirable sensory properties of the studied products.

## 2. Materials and Methods

### 2.1. Sausage Samples

Five types of dry-fermented sausages were collected at the international sausage fairs held in Croatia during 2018 and 2019. Sausages were produced in 2017–2019 timeframe by forty-seven Croatian family farms seated in two different Croatian regions: eastern Slavonia (Kulenova Seka, *n* = 24; Slavonian domestic sausage, *n* = 27) and western Istria (occupying the northern Croatian coast) (Istrian domestic sausages, *n* = 36; Kosnica, *n* = 21; Istrian-type sausage with additional ingredients, *n* = 33). A total of 141 dry-fermented sausages were sampled, out of which each sample was taken in three replicates and in the amount of 1.5 to 2 kg.

According to the Regulation [22], these products belong to the group of dry-fermented sausages enlisted within the category of nonthermally processed meat products. They are produced according to traditional recipes, from pork meat of the first and the second category and fat with the addition of salt and different spices stuffed into casings. Prior to stuffing, meat and fat are mechanically minced. Spices used in the production of the analyzed dry-fermented sausages contribute to the aroma of the products and are mentioned in Table 1. After stuffing, sausages undergo drying and ripening with or without smoking. Smoking is characteristic only for Slavonian domestic sausage and Kulenova Seka (i.e., sausages produced in the eastern Croatia). Ripening is the most demanding production operation and has a huge impact on physicochemical and sensorial properties of the final product. Ripening environment and duration depend on the type of product. Detailed information on every type of sausage under study is given in Table 1. Slavonian domestic sausage is protected by geographic indication (PGI) under the EU legislation, while Istrian domestic sausage pends such a designation in an ongoing procedure.

### 2.2. Isolation and Identification of Superficial Mycobiota

Molds present on the surface of analyzed traditional dry-fermented sausages were isolated and identified using a traditional method of depiction of macroscopic and microscopic morphologic characteristics and corroborated using a molecular method described earlier by Zadravec et al. [19]. Deoxyribonucleic acid (DNA) was extracted from about 100 µg of mold colonies using the DNeasy Plant Mini Kit (Qiagen, Hilden, Germany) according to manufacturer instructions. Primers specific for internal transcribed spacer (ITS)—ITS1 (5′-TCCGTAGGTGAACCTGCGG-3′) and ITS4 (5′-TCCTCCGTCTATTGATATGC-3′), beta-tubulin (benA)—Bt2a (5′-GGTAACCAAATCGGTGCTGCTTTC-3′) and Bt2b (5′-ACCCTCAGTGTAGTGACCCTTGGC-3′) & calmodulin (CaM) loci—Cmd5 (5′-CCGAGTACAAGGARGCCTTC-3′) and Cmd6 (5′-CCGATRGAGGTCATRACGTGG-3′) were selected for polymerase chain reaction (PCR )() amplification according to traditionally identified mold genus. PCR was performed in a total reaction volume of 25 µL as described by Pleadin et al. [9], under the following cycling conditions: 95 °C for 5 min followed by 40 cycles at 94 °C for 30 s, 56 °C for 30 s, 72 °C for 60 s, concluding with 72 °C for 10 min. The PCR products were checked using gel electrophoresis in 1.5%-agarose gel and visualized by virtue of ultraviolet (UV)trans-illumination. All PCR products of an adequate size were purified prior to sequencing using an ExoSAP-IT PCR cleanup reagent (Affymetrix, Santa Clara, CA, USA). Purified samples were sent to the Macrogen, Inc. (Amsterdam, the Netherlands) for paired-end sequencing. The obtained sequences were aligned using the Lasergene DNASTAR 17 (Madison, WI, USA). Edited sequences were compared to those available from the GenBank using the BLAST algorithm (http://blast.ncbi.nlm.nih.gov/Blast.cgi). Obtained sequences are deposited in GenBank database with accession numbers as follows: MT584832 for *M. racemosus*, MT584831 for *A. flavus*, MT627166 for *P. commune*, MT664968 for *A. proliferans*, MT664969 for *A. pseudoglaucus*, MT664970 for *P. solitum*, MT664971 for *P. nalgiovense*, MT664972 for *P. citrinum*, MT664973 for *P. salamii*, MT664974 for *P. polonicum* and MT664975 for *A. niger*.

### 2.3. Sensory Evaluation

Sensory evaluation of the sausage samples was conducted by a trained panel (of 9 assessors). The assessors were selected and generically trained according to the ISO standard [25]. Sensory analysis was carried out in a sensory laboratory of the Faculty of Food Technology and Biotechnology University of Zagreb according to the ISO standard [26] (room technical requirements: relative humidity 50%–55% and temperature 20–22 °C). Sensory evaluation made use of a quantitative descriptive analysis (QDA) based on the numeric and unipolar intensity scale developed in collaboration with the Centro Studi Assaggiatori (Brescia, Italy). The intensity of each sensory property was estimated using a numeric scale calibrated from left to right, with “0” indicating the absence of a given sensory property and “9” indicating its strongest intensity. On each panel session, a repeat sample of each product group was provided; for each assessor, a number of statistical parameters descriptive of his/her efficacy was calculated.

Individually coded samples were served at room temperature (3 slices of 2-mm thickness) in sensor compartments together with all materials necessary for sensory evaluation. Each panelist evaluated the intensity of both objective and subjective product properties. Sensorial assessment of objective product qualities embraced the assessment of visual qualities (color of the minced meat, color uniformity, visible fat content, cohesiveness), olfactory qualities (favorable smell, unfavorable smell, smoky smell), mouth feel (tenderness, juiciness, saltiness, sweetness, sourness, bitterness, spiciness, hotness) and aroma (generated by aromatic herbs, spice herbs, ripen meat, biochemical product properties, fresh pork meat, molds). The assessment of subjective product properties embraced the assessment of cross-section attractiveness, smell attractiveness, consistency attractiveness, taste attractiveness, product maturity, richness of favorable aromas, steadiness of favorable aromas and overall product attractiveness.

### 2.4. Determination of Colorimetric Parameters

Color parameters were determined in homogenized samples using a Chroma Meter CR-400 SET colorimeter (Konica Minolta, Tokio, Japan) according to manufacturer instructions and the CIE (Commision Internationale de l’Eclairage) L* a* b* color system that includes the measurements of lightness (L*), redness (a) and yellowness (b*). L* is the measure of lightness and spans from “0” (black) to 100 (white). Highly positive a* values indicate redness, while highly negative values indicate greenness. b* values indicate yellowness to blueness. Before the analysis, the instrument was white-calibrated (D65; Y = 85.6, x = 0.3183, y = 0.3357).

### 2.5. Determination of Physical and Chemical Properties

Samples were homogenized using a Grindomix GM 200 homogenizer (Retsch, Haan, Germany). Chemical properties of the sausages were determined using standard analytical methods foreseen for the analysis of water [27], ash [28], fat [29] and total protein content [30]; the aforementioned made use of an UF75 Plus oven (Memmert, Schwabach, Germany), a LV9/11/P320 furnace (Nabertherm, Lilienthal, Germany), a Soxtherm 416 automated device (Gerhardt, Konigswinter, Germany) and a Vapodest 50 s automated distillation & titration device (Gerhardt, Konigswinter, Germany). Carbohydrate content was calculated based on the parameters detailed above. Quality control was performed using the TET003RM Reference Material (Fapas, York, UK).

Sodium chloride content was determined stoichiometrically based on the sodium content measured using the in-house validated potentiometric method and an Easy Na analyzer (Mettler Toledo, Schwerzenbach, Switzerland). Quality control was performed using the TO1124QC Reference material (Fapas, York, UK).

The pH value was measured using a digital Seven Compact pH meter (Mettler Toledo, Schwerzenbach, Switzerland), while water activity (aw) was determined using a HygroPalm AW (Rotronic Emin Tech., Lund, Sweden), in both cases in full line with manufacturer instructions.

All chemicals used for the analyses were of an analytical grade. The results are expressed as the mean value obtained from two parallel runs.

### 2.6. Determination of Fatty Acids

Sample preparation for the analysis of fatty acid methyl esters was performed according to the ISO standard [31] described earlier by Pleadin et al. [32]. Fatty acid methyl esters were analyzed using a 7890B gas chromatographer equipped with flame ionization detector (FID) (Agilent Technologies, Santa Clara, CA, USA) and DB-23 capillary column (60 m, 0.25 ID, 0.25 μm) according to the standard analytical protocol [33] with detailed conditions described earlier by Pleadin et al. [32].

Fatty acid methyl esters were identified by comparing their retention times with those of fatty acid methyl esters contained by the standard mixture (Supelco^TM^ 37 Component FAME Mix, Bellefonte, PA, USA). Verification of the method was described earlier by Pleadin et al. [34]. The results are expressed as the percentage (%) of a given fatty acid in the total fatty acid share.

### 2.7. Fat Quality Indices

The results concerning the fatty acid composition were used for the calculation of fat quality indices using the following formulas presented in Equations (1)–(3) [35,36]:The atherogenic index (AI) = [12:0 + (4 × 14:0) + 16:0]/(the sum of monounsaturated fatty acids (MUFA) + PUFA n-6 + PUFA n-3)(1)
The thrombogenic index (TI) = (14:0 + 16:0 + 18:0)/[(0.5 x the sum of MUFA + 0.5 × PUFA n-6 + 3 × PUFA n-3) + (PUFA n-3/PUFA n-6)(2)
The ratio of hypocholesterolemic over hypercholesterolemic fatty acids (H/H) = (C18:1n-9 + C18:2n-6 + C20:4n-6 + C18:3n-3 + C20:5n-3 + C22:5n-3 + C22:6n-3)/(C14:0 + C16:0)(3)

### 2.8. Statistical Analysis

Statistical analyses were performed using the SPSS Statistics Software 22.0 (IBM, New York, NY, USA) and the Big Sensory Soft (Centro Studi Assaggiatori, Brescia, Italy). The results were tested for the normality of their distribution using the Shapiro-Wilks test. In order to determine the statistical significance of the differences in chemical, physical, colorimetric, sensory and fatty acid parameters among the analyzed sausages, the one-way ANOVA test was used, while for mycobiota, the Kruskal–Wallis test was used. Mann–Whitney U test was used to determine the statistical significance of differences in number of identified mold isolates among smoked and non-smoked sausages coming from two different regions. A Pearson’s multivariate correlation analysis was performed to determine the relationships among fat, protein, water and color parameters. Decisions on statistical relevance were made at the significance level of *p* ≤ 0.05.

## 3. Results and Discussion

### 3.1. Superficial Mycobiota

Different factors affect mold growth on the surface of dry-fermented sausages, the major being aw, temperature, pH-value and salt content [37,38]. Dry-fermented sausages are characterized by low pH and aw values and high salt content, as well as low and moderate ripening temperatures, all of that facilitating the growth of xerotolerant and xerophilic molds, such as those of the *Penicillium* and the *Aspergillus* genera [8,17]. The *Penicillium* species prefers to grow at aw< 0.80 and temperatures at which sausages are ripened, while the *Aspergillus* species prefers to grow at aw< 0.90 and is usually present in subtropical and tropical areas. The *Mucor* species prefers low temperatures and is often isolated from sausages ripened during winter [19].

Mold species identified on the surface of analyzed Croatian traditional dry-fermented sausages are shown in Table 2. As apparent, 71% of the identified molds were of the *Penicillium*, 18% of the *Mucor* and 11% of the *Aspergillus* genus. The percentage of *Mucor* genus isolates wasn’t significantly different from the percentage of isolates of other mold genera, while the *Penicillium* genus molds were present in a significantly higher percentage of isolates than those of the *Aspergillus* genus (*p* = 0.033), showing a greater species diversity, since six *Penicillium* and four *Aspergillus* species were identified. The most represented species (given with the pertaining relative densities, Dr% = the number of isolates of a given species/total number of fungi isolated × 100) of the *Penicillium* genus were *P. nalgiovense* (28%), *P. solitum* (23%) and *P. commune* (15%), while the dominating *Aspergillus* genus mold was *A. pseudoglaucus* (6%). *Mucor racemosus* (Dr =18%) was the only species detected among the *Mucor* genus. As for the *Penicillium* genus, the results of this study are consistent with earlier data on the predominance of *P. nalgiovense* in the sausage surface mycoflora, followed *P. olsonii, P. chysogenum, P. commune* and *P. solitum* [8,15,16,18]. In the research of Vila et al. [18] performed on Argentinean dry-fermented sausages, the *Mucor* was also the second most frequently isolated mold genus, with *M. racemosus* being the predominant species, while the *Aspergillus* genus was the least represented.

No significant difference in the percentage of mold isolates among five different sausages types (*p* = 0.655), as well as between smoked and non-smoked sausages that came from different regions—eastern Slavonia with continental climate and western Istria with moderate climate (*p* = 0.250) were found. In the research by Zadravec et al. [19], smoked and non-smoked dry-cured meat products also failed to show significant difference, while products coming from different regions-western and eastern Croatia-differed significantly. Smoking can affect mycobiota since the reduction of *P. solitum* and *P. commune* in smoked products was observed [39]. In this study, a lower occurrence of *P. solitum,* but not *P. commune* in smoked than non-smoked sausages was also observed, as well. Sausages coming from eastern Croatia showed greater species diversity (7–8 species) than those coming from western Croatia (5–7 species), presumably because the first are ripened (up to a month) longer.

Some of the mold species can produce mycotoxins that can have different hazard effects in human body and jeopardize consumer health [9,17,19]. Out of molds identified in this study, potential mycotoxigenic species are *Penicillium commune* as a cyclopiazonic acid (CPA) producer, *Penicillium citrinum* as a citrinin (CIT) producer and *P. polonicum* as a verrucosidin & nephrotoxic glycopeptides producer, *Aspergillus flavus* as an aflatoxin (AF) & CPA producer and *Aspergillus niger* as an ochratoxin A (OTA) producer. The highest number of mycotoxigenic species was isolated from Slavonian domestic sausage (12), followed by Kulenova Seka (6). Their responsibility for AF and OTA contamination was evidenced in earlier studies of Croatian traditional dry-cured meat products [9,19].

### 3.2. Sensory Profile

The acceptability of dry-cured meat products is in dependence of their sensory characteristics, including color, aroma, flavor, texture, aftertaste and overall appearance. Differences in the above allow for the mutual distinction of various products and it is therefore worth the effort to identify and quantify these sensory quality attributes [40]. In addition, these traditional dry-fermented sausages have not been previously characterized and compared based on their nutritional and sensory properties in relation to food quality and consumer acceptability, as well as their impact on consumer health. The appropriate attributes have been selected and applied through complex procedures involving sensory vocabulary generation and generation of lexicon to be used to describe fermented meat products, as well as quantitative–descriptive analysis [40].

The results of sensory evaluation conducted using quantitative descriptive analysis of Croatian dry-fermented sausages are shown in Figure 1 and Figure 2. Statistical analysis found significant differences (*p* < 0.05) in 16 out of 21 attributes in all categories, i.e., visual qualities, odor, texture, taste and aroma. No statistically significant differences were found among subjective sensorial parameters of dry-fermented sausages under study.

Significant variations were found in visible fat content and cohesiveness, where the visible fat content was higher and the cohesiveness lower in Kulenova Seka in comparison with Istrian domestic sausage and Istrian-type sausage with additional ingredients. Properties such as cohesiveness and tenderness can be related to pH-value and fat & salt content of meat products, so that the significant difference in pH and fat values, determined among different types of sausages in this study, could explain this variability [41,42]. In dry-fermented sausages, texture is also associated with water content, which is significantly higher in Kulenova Seka than other sausages under study which can also be the reason that Kulenova Seka was characterized as significantly tender in comparison with Istrian domestic sausage.

Smoky smell was reported only for Slavonian domestic sausage and Kulenova Seka, which was to be expected since smoking is a part of processing of only these sausages produced in the eastern Croatia. In general, unfavorable smell was rarely reported by assessors for any of the sausages under assessment (score: 0.7–1.5), however significantly more frequently for Istrian domestic sausage than other sausage types. On the surface of the latter, the highest number of *Mucor* isolates was found, which can be responsible for the ammoniacal and other off-flavors and represent a contamination indicator. On top of that, the presence of *A. pseudoglaucus* responsible for an off-taste and unwanted color of the product, was established, as well [8,43]. *P. commune* that was isolated from all sausage types can also be responsible for an unfavorable sensory profile, such as phenol acid defect (off-odor) earlier detected in some hams during ripening [16].

Molds influence the sausage taste and aroma mainly due to their ability to produce lipid– and protein-degrading enzymes, where correlation among sensory properties and proteolytic and lipolytic mold activity was established. Studies of the impact of surface molds on sausage aroma also show that molds of the *Penicillium* and the *Aspergillus* genera are able to produce odor-active compounds, such as ketones and alcohols [7,10,44]. In general, all sausages analyzed in this study showed good acceptability, with values around 7 out of possible 9, and had a similar smell & taste attractiveness and aroma richness, as well as a favorable smell. In all of them, *P. nalgiovense, P. solitum* and *P. commune*, which can have desirable effects on sensorial properties, were isolated very frequently. As for the surface molds-related overall appearance of the final products, white or grayish fungal coat was approved, while brown/green sporulation and black spot formation were considered to be spoilage signs [8]. *P. nalgiovense* gives a sausage a whitish/gray coloration approved by consumers and contributes to flavor and improved texture, so that it is usually selected as a commercial starter culture, although some strains can produce penicillin that should not be present in food [15,16]. In comparison with *Mucor*, commercial *P. nalgiovense* has a superior appearance, but a lower enzymatic activity [10]. *P. solitum* can grow at refrigeration temperatures and is known to produce lipases, proteases and compactins [7,16]. In cheeses, *P. commune* was evidenced to contribute to changes during ripening and flavor, but *P. commune* isolated from dry-cured ham was also shown to have a vivid proteolytic and lipolytic activity [45,46]. *P. salamii* deemed to produce a favorable surface coverage, appearance and flavor of sausages and no mycotoxins [8], was isolated in this study only in one sample of Istrian- type sausage with additional ingredients.

Significant differences were found in salty, sweet and sour taste. Salinity of the products was characterized as moderate (4–5.5), with low values for sweet (1.7–2.8), sour (0.9–1.8) and bitter taste (around 1). Kosnica was assessed to be significantly saltier and more acidic than other sausages. Istrian domestic sausage was shown to be statistically significantly sweeter and less salty than Kulenova Seka. Dry-fermented Istrian domestic sausage must have a delicate, but never sour taste and moderate salinity, while, according to the specification, Slavonian domestic dry-fermented sausages should be mildly hot, but never bitter [23]. Slavonian domestic sausage and Kulenova Seka are hotter (5.53% and 5.69%, respectively) due to hot red pepper added in their stuffing than other sausages under study that are from the western Croatia, whose values in this regard were close to 0. For Kulenova Seka and Slavonian domestic sausage is also shown to contain less aromatic and more spicy herbs than other sausages. The results are consistent with the sausage recipes (spices added) considering their geographic origin. Moldy and fresh pork meat aroma attributes generally scored very low (<1 and 0.4–1.2, respectively). Ripen meat aroma is significantly more pronounced in longer-ripened Slavonian and Special sausages than in Istrian sausage.

### 3.3. Color Parameters

The color of a final meat product largely influences consumer preferences and is the main aspect of the product quality, so that a product may be rejected simply because of its color even before other properties get to be evaluated. As a quality parameter, color has been widely studied in fresh meat and cooked products, but dry-cured meat products have received less attention because in this type of products color formation takes place during different processing stages. In case of dry-fermented sausages, color depends on a number of factors, such as the composition of the sausage, the fat-lean ratio, the quantity and the type of spices and additives and technological operations applied [47,48].

The results of color measurement obtained in this research, together with the associated *p*-values, are shown in Table 3. Sausage samples significantly differed in all measured color parameters (L*, a* and b*), but the results are in accordance with those found in the literature for similar product types [4,24,42]. Similar L*, a* and b* values (32.84–37.34, 6.45–10.83 and 3.80–7.55, respectively) were reported for Istrian sausage by Kovačević [49]. Values obtained for Slavonian domestic sausage and Kulenova Seka are in correlation with the values determined in the research of Slavonski Kulen by Kovačević et al. [42]. Slavonian Kulen is the sixth most common traditional Croatian dry-fermented sausage and the first designated as protected by geographic indication (PGI) but was not included in this research because its sensory and nutritional properties were already investigated [24,42]. Lightness (L*) was significantly higher in Kosnica in comparison with Istrian-type sausage with additional ingredients, in which this value was the lowest. Redness (a*) and yellowness (b*) were significantly higher in Slavonian domestic sausage and Kulenova Seka in comparison with other sausages. The above result could probably be attributed to sweet and hot red pepper used only in the production of Kulenova Seka and Slavonian domestic sausage. Yellowness is probably also related to red pepper, which is to say, to yellow carotenoids coming from it; the same was shown for similar products, such as domestic Slavonian Kulen, Spanish dry-fermented sausage Chorizo de Pamplona and Portuguese Iberian sausage [4,41,42]. In this study, a significant inverse correlation was found among b* values and protein content for sausages with red pepper added, Slavonian domestic sausages and Kulenova Seka (*r* = −0.69 and *r* = −0.78), as well as among a* values and protein content, but only for Slavonian domestic sausage (*r* = −0.78). Research has shown that the lower the protein content, the smaller the a* value due to the dilution of myoglobin as a consequence of reduced protein content [41], but results of this research can be probably attributed to addition of pepper for which was shown to reduce L*, whereas a* and b* values increase [48]. In general, low fat and high water content lead to higher a* and lower L* values. No significant correlations were found between color parameters and water content, while a positive correlation was determined among L* values and fat content for Kosnica (*r* = 0.81).

### 3.4. Physical and Chemical Properties

Physicochemical and sensory characteristics differ depending on product composition, processing procedures and complex protein, lipid and carbohydrate metabolism coming as a result of meat bacteria and mold-induced enzymatic activity and oxidation processes [14]. The results of analyses of physical and chemical parameters descriptive of traditional dry-fermented sausages under this study are shown in Table 4. Traditional sausages significantly differ in pH and aw values, as well as in fat, water and protein content (*p* < 0.05). No difference was observed in ash, carbohydrate and salt content (*p* > 0.05).

During fermentation and ripening of dry-fermented sausages, a decrease in pH and aw value, necessary to achieve product stability and safety, occurs [24]. The aw values for all sausages under study were lower than 0.85. pH-value is an indicator of fermentation and ripening and mainly comes as a result of the lactic acid bacteria activity [1]. The lowest pH value of 5.23 was measured in Kulenova Seka, because this sausage ripens the longest than Istrian sausage in which the highest pH value of 6.26 was measured. The obtained values are typical of low-acidic meat products (final pH 5.3–6.2) [4]. pH and aw values of traditional fermented sausages from other Mediterranean countries and those of the Croatian dry-fermented sausage domestic Slavonian Kulen were in range of 5.06–6.3 for pH and 0.82–0.92 for aw [1,4,12,13,34].

Generally, sausage production technology leads to the decrease in water content and increase in ash, protein and total fat content [24]. The highest water content was measured in Kulenova Seka (34.05%) accordant with maximum value allowed (40%) and the lowest in Slavonian domestic sausage (22.07%). The main property that distinguishes Slavonian domestic sausage from other similar European dry sausages is lower moisture content [50]. Protein content was highest in Istrian-type sausage with additional ingredients (35.09%) and lowest in Kosnica (28.79%). It can be seen that these products are rich in proteins, which makes them high-quality products. Protein content of other dry-fermented sausages found in literature spans from 18.7% to 43.2% [1,4,12,34], these values most commonly approximating to 30%–40%. The ratio of water over proteins in the analyzed sausages was 0.63 in Istrian-type sausage with additional ingredients to 1.03 in Kulenova Seka. Moisture/protein ratio can serve as the basis for classification of sausages into dry-fermented, should its value be below 1.5 [51].

Fat is considered to be the most variable ingredient in meat products, whose quantity and quality influence not just nutritional value, but also organoleptic properties and health implications of the product. The fat content depends on the choice of raw materials (for example fattier or leaner meat), the recipe and the production technology [34,52]. In Croatian sausages, on top of intramuscular fat already present in the muscle tissue, adipose tissue is added into stuffing in the amount of 25 do 30%, the baseline amount which, when it comes to fermented sausages, ranges from 40% to 45%, thereby being further increased during drying [41]. In this study, the fat content was significantly lower in Kulenova Seka (27%) in comparison with other sausage types, with average values of 37%–41%. Fat content in the amount of 30%–50% was found in sausages from other Mediterranean countries, such as those from Italy, Portugal and Spain [1,4,41]. In the research of Croatian Slavonian Kulen, fat values were found to be lower than 30% (14%–29%) [42].

Salt (sodium chloride) improves organoleptic properties of meat products due to its ability to enhance product texture, color and taste, but also preserves them due to the inhibition of activities of various microorganisms [5]. Generally, in the stuffing of fermented sausages the average salt content ranges from 2.0% to 2.6% and, due to the water loss during drying, this proportion in the finished product falls to 3.3%–5.5% [24]. Salt contents in typical Sicilian salami and Portuguese traditional sausages Catalao and Salsichao were almost 2× higher, with values of 6% [5,53]. Optimal reduction of salt content should be considered as a health advantage, but not at the expense of product safety, quality and sensory properties. For Slavonian domestic sausage, an optimal reduction of salt content in final products that does not affect sensory, microbiologic and physicochemical properties was shown to be roughly 3%, as also determined for Portuguese traditional sausages Catalao and Salsichao [5,6]. All sausages analyzed in this study had a similar salt content of around 3.5%, which enlists them into the category of less salty products, given the average sausage salt content of 3%–6% [14,24].

The determined carbohydrate content (of 0.17%–0.56%) can be explained by the sausage preparation process, since in stuffing of some of the traditional dry-fermented sausages sugars can be added in order to accelerate fermentation, although this practice can be far more frequently encountered in industrial than in household production settings. For example, in Slavonian Kulen sugars are usually added in the amount of 0.4% [49].

### 3.5. Fatty Acid Composition and Lipid Quality Indices

Fatty acid (FA) composition of the analyzed traditional Croatian dry-fermented sausages and fat quality indices calculated based on the latter composition are shown in Table 5. FA most commonly determined in all analyzed sausages were MUFAs, followed by saturated fatty acids (SFAs) and PUFAs. Studies have shown the average fatty acid meat content to be around 40% for SFAs, 40% for MUFAs and about 2%–25% for PUFAs [52]. The most common individual fatty acids in this study were oleic (18:1n-9c), palmitic (16:0), stearic (18:0) and linoleic (C18:2n-6c) acid. Oleic acid is the fatty acid most represented in pig fat (45%–50% of all FAs). In the sausages analyzed in this study, oleic acid accounted for 85%–86% of all MUFAs. As for SFAs, palmitic and stearic acids represented up to 94% of them all. The dominance of n-6 PUFAs over n-3 PUFAs is primarily attributable to linoleic acid, which accounted for 86%–93% of total PUFAs.

In general, sausages of various types significantly differed in their fatty acid composition. As for individual fatty acids, differences were found in the representation of 13 fatty acids (C10:0, C14:0, C15:0, C16:0, C16:1t, C18:0, C18:1n-9t, C18:1n-9c, C18:3n-3, C20:1n-9, C20:2n-6, C20:4n-6, C23:0). Fatty acids have been shown not to depend on processing conditions such as ripening, but rather on the composition of raw materials [5,53]. Differences were also found in the SFA and MUFA share, where Kulenova Seka had a lower SFA and higher MUFA share in comparison with Kosnica and Istrian-type sausage with additional ingredients. A lower SFA percentage is mainly explainable by the significantly lower palmitic and stearic acid percentage, while the higher MUFA percentage is to be attributed to the higher oleic and eicosenoic (C20:1n-9) acid percentage. When it comes to PUFAs, significant difference was found only in the representation of n-3 fatty acids, where Slavonian domestic sausage had a significantly higher PUFA share than Kosnica. Kosnica had the lowest C18:3n-3 share. Out of n-3 fatty acids, only 1–2 were identified (C18:3n-3, C20:3n-3), depending on the sausage type. No differences in linoleic acid (C18:2n-6c) and n-6 fatty acid share were observed.

*Trans*-fatty acids (TFA) have an adverse impact on cholesterol metabolism and are linked to an increased risk of the development of cardiovascular diseases. Health organizations have recommended the reduction of TFA intake to the lowest level possible, while the representation of industrial *trans*-fatty acids in final products should not exceed 2/100 g of fat [54]. Ruminant meat contains *trans*-fats generated by biohydrogenation that takes place in the rumen, but these fats can also be present in the meat coming from non-ruminants, such as pigs, however mostly in low concentrations and due to feeding on trans-fats-containing feed [55]. The representation of total TFAs in pork meat was established to be 0.2%–2.2%. The study of the Spanish dry-fermented sausage Chorizo de Pamplona conducted by Muguerza et al. [56] came up with the total share of TFAs in total FAs of 1.07%. In this study, the share of analyzed TFAs was lower than 1% in all sausages under study, their representation thereby being statistically significantly lower in Slavonian domestic sausage than Istrian-type sausage with additional ingredients.

Modern diets are characterized by an excessive intake of fat, especially SFAs and by the disturbed balance in terms of an increased n-6 PUFA in comparison with n-3 PUFA intake. Therefore, PUFA/SFA and n-6/n-3 ratios are the most common parameters used to evaluate the nutritional quality of fat and potential effects on consumer health [52,57]. However, when it comes to consumer health issues, literature also suggests indices which take functional properties of fatty acids into account, such as IA, IT and H/H. IT is defined as the relationship between pro-thrombogenic (saturated) and anti-thrombogenic (unsaturated) FAs and shows the tendency to blood vessel clotting. IA indicates the relationship between the sum of the main saturates and the main non-saturates, the former being considered pro-atherogenic and the latter being considered anti-atherogenic [35]. H/H is the ratio between hypocholesterolemic and hypercholesterolemic fatty acids that take into account known effects of certain fatty acids on cholesterol metabolism [36]. In order to meet health recommendations or reduce the risk of developing cardiovascular and other chronic diseases, the n-6/n-3 ratio should not exceed 4 and the PUFA/SFA ratio should be greater than 0.4 [57]. Values recommended for IA and IT are lower than 1, while the recommended H/H is higher [32].

In sausages analyzed within this frame, statistically significant differences were found in all analyzed parameters except for the PUFA/SFA ratio. This ratio (0.2–0.3) was similar for all sausage types and was below the recommended value, but in line with the literature data for other dry-fermented sausage types, reported to span from 0.2 to 0.4 [4,52,58,59]. Significantly lower n-6/n-3 ratio was determined in Istrian-type sausage with additional ingredients and Slavonian domestic sausages in comparison with Kosnica and Istrian domestic sausage, but all sausage types studied had n-6/n-3 ratios higher than those suggested by international health organizations. These values are in agreement with those claimed by Jimenez-Colmenero [60], who reported that data from studies on meat products show that n-3 PUFAs are present in these products in poor quantities, hence, contribute little to dietary recommendations fulfilment. Undesirable ratios in disagreement with healthy diet recommendations, ranging from 7.6 to 21, were also determined in other studies of dry-fermented sausages [34,52,59,60], whereas for sausages analyzed in this research the results are at the upper limits, i.e., 4–5 times higher than recommended, indicating the need for fat reformulation.

Kulenova Seka had a significantly lower IA and a significantly higher H/H than other analyzed sausages, as well as a higher IT value than Kosnica, showing the most representable values according to recommendations. When it comes to IA and H/H, other sausage types also meet the recommendations, while the obtained IT values are slightly higher than the recommended value of 1, but in line with literature data on similar sausage types (1.09–1.35) [58,61]. H/H value around 2 is characteristic of pork meat and pork meat products [52].

## 4. Conclusions

The results did not indicate any significant differences in the number of mold isolates among different sausage types, as well as among smoked and non-smoked sausages coming from two different regions, although sausages coming from eastern Croatia exhibited a greater species diversity. The highest number of isolated mycotoxigenic species was found in Slavonian domestic sausage. Different recipes used in the production of the investigated Croatian dry-fermented sausages and different conditions under which they get to be processed, result in significant variations in their nutritional composition in terms of fat, protein and water content and fatty acid profile, as well as in terms of their sensory properties. Regardless of the aforementioned, good acceptability was shown for all sausage types. According to health recommendations, Kulenova Seka showed the most representable values in terms of fat quality indices (IT, IA and HH), while for other sausage types under study these indices were not within the recommended values but were characteristic of this type of meat products. The results suggest the need for certain modifications of fat and fatty acid composition and, to a lesser extent, of salt content, so as to reduce the unwanted health implications, but not at the expense of product safety, quality and acceptability. Further studies of traditional dry-fermented sausages in terms of their safety and the occurrence of unexplored mycotoxins should be performed.

## Figures and Tables

**Figure 1 foods-09-00990-f001:**
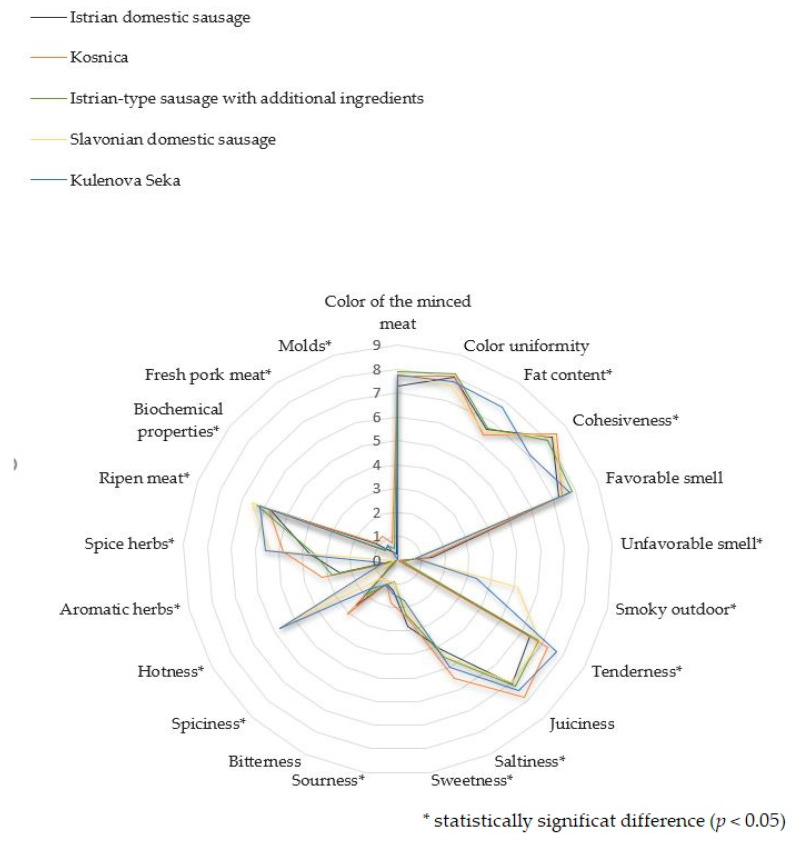
Objective sensorial characteristics of Croatian traditional sausages (appearance, odor, texture, taste and aroma).

**Figure 2 foods-09-00990-f002:**
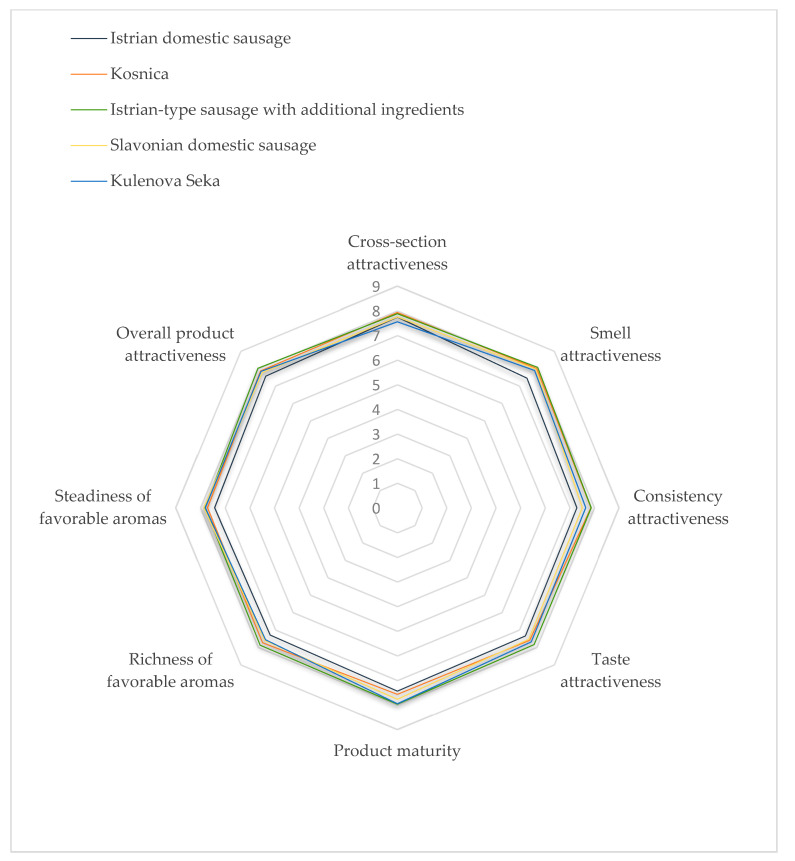
Subjective sensorial characteristics of Croatian traditional sausages.

**Table 1 foods-09-00990-t001:** Specificity of the production technology and properties of the analyzed dry-fermented sausages [22,23,24].

Type of Sausage	Perforation Diameter (mm)	Spices	Smoking (days)	Ripening Time (months)	Ripening Conditions	Physicochemical Properties	Croatian Region
Istrian domestic sausage	10–12	Kitchen salt, pepper, cooked Malvasia and garlic	Non-smoked	1.5–2	Temp. 9–16 °C; relative humidity 65%–85%	Fat < 40%; Proteins > 16%	Istria and Primorje—Western
Kosnica	>12	Kitchen salt, pepper, cooked Malvasia and garlic	Non-smoked	2–2.5	Temp. 9–16 °C; relative humidity 65%–85%	Fat < 40%; Proteins > 16%	Istria and Primorje—Western
Istrian-type sausage with additional ingredients	10–12	Not specified	Non-smoked	1.5–2	Temp. 9–16 °C; relative humidity 65%–85%	Fat < 40%; Proteins > 16%	Istria and Primorje—Western
Slavonian domestic sausage	6–8	Kitchen salt, sweet and hot red pepper, garlic	14	2–2.5	Temp. ≤16 °C; relative humidity 70%–85%	Fat < 40%; aw < 0,90; Proteins > 16%	Slavonia and Baranja—Eastern
Kulenova Seka	6–8	Kitchen salt, sweet and hot red pepper, garlic	14	2.5–3	Temp. 14–17 °C; relative humidity 70%–80%	Fat < 40%; Proteins > 16%	Slavonia and Baranja—Eastern

**Table 2 foods-09-00990-t002:** Superficial mycobiota found on Croatian traditional dry-fermented sausages.

Identified Molds	Percentage of Mold Isolates (%) *				
	Istrian Domestic Sausage	Kosnica	Istrian-Type Sausage **	Slavonian Domestic Sausage	Kulenova Seka
*P. solitum*	7.69	5.13	5.13	2.56	2.56
*P. nalgiovense*	5.13	4.27	8.55	6.84	3.42
*P. citrinum*	0.85			2.56	
*P. commune*	3.42	0.85	1.71	5.13	3.42
*P. salamii*			0.85		
*P. polonicum*					0.85
*Penicillium* sp.	17.09	10.26	16.24	17.09	10.26
*A. proliferans*	0.85	0.85			
*A. pseudoglaucus*	0.85		1.71	2.56	0.85
*A. niger*				0.85	
*A. flavus*				1.71	0.85
*Aspergillus* sp.	1.71	0.85	1.71	5.13	1.71
*M. racemosus*	5.13	0.85	5.98	1.71	4.27
*Mucor* sp.	5.13	0.85	5.98	1.71	4.27
Total percentage^a^	23.93	11.96	23.93	23.93	16.23

* Number of isolates of a species or genus/total number of isolates × 100; ** Istrian-type sausage with additional ingredients; ^a^ The total percentage of isolated molds in a given sausage type; Results are statistically evaluated (*p* < 0.05) according to the type of sausages (*p* = 0.655), mold genus (*p* = 0.033) and regions of origin (*p* = 0.250).

**Table 3 foods-09-00990-t003:** Color (L* a* b* values) of Croatian traditional dry-fermented sausages.

Type of Sausage	No of Samples	L*	a*	b*
Istrian domestic sausage	36	39.21 ± 2.15 ^ab^	6.03 ± 1.25 ^b^	7.13 ± 0.76 ^b^
Kosnica	21	43.94 ± 4.05 ^a^	8.59 ± 1.62 ^b^	7.92 ± 1.15 ^b^
Istrian-type sausage with additional ingredients	33	38.38 ± 4.11 ^b^	8.60 ± 2.63 ^b^	6.77 ± 1.40 ^b^
Slavonian domestic sausage	27	40.26 ± 3.01 ^ab^	15.84 ± 3.40 ^a^	19.07 ± 5.84 ^a^
Kulenova Seka	24	39.03 ± 2.11 ^ab^	15.89 ± 3.25 ^a^	16.95 ± 2.89 ^a^
*p*-value		0.010	0.000	0.000

L^*^= lightness, a^*^ = redness, b^*^ = yellowness; ^a,b^ values within a column with no common superscript differ significantly (p < 0.05); p-value refers to results of analyzed parameter per column among five sausage types.

**Table 4 foods-09-00990-t004:** Chemical and physical properties of Croatian traditional dry-fermented sausages.

Parameters	Istrian Domestic Sausage (*n* = 36)	Kosnica (*n* = 21)	Istrian-Type Sausage ** (*n* = 33)	Slavonian Domestic Sausage (*n* = 27)	Kulenova Seka (*n* = 24)	*p*-Value
a_w_	0.83 ± 0.03 ^ab^	0.83 ± 0.04 ^ab^	0.80 ± 0.03 ^b^	0.79 ± 0.03 ^b^	0.85 ± 0.02 ^a^	0.002
pH	6.26 ± 0.47 ^a^	5.90 ± 0.89 ^ab^	5.94 ± 0.65 ^ab^	5.32 ± 0.37 ^ab^	5.23 ± 0.32 ^b^	0.001
Water *	23.74 ± 3.60 ^b^	27.83 ± 3.41 ^ab^	22.15 ± 4.25 ^b^	22.07 ± 2.52 ^b^	34.05 ± 4.66 ^a^	0.000
Ash *	5.10 ± 0.63	5.39 ± 0.92	5.40 ± 0.35	5.31 ± 0.76	5.44 ± 0.95	0.813
Fat *	38.27 ± 4.69 ^a^	37.44 ± 5.97 ^a^	36.85 ± 4.65 ^a^	40.71 ± 5.39 ^a^	27.30 ± 4.80 ^b^	0.000
Proteins *	32.59 ± 2.15 ^ab^	28.79 ± 3.18 ^b^	35.09 ± 4.05 ^a^	31.36 ± 4.45 ^ab^	33.05 ± 1.19 ^ab^	0.005
Carbohydrate *	0.31 ± 0.65	0.55 ± 0.77	0.51 ± 0.91	0.56 ± 1.10	0.17 ± 0.30	0.804
Salt *	3.39 ± 0.76	3.61 ± 0.80	3.56 ± 0.42	3.43 ± 0.52	3.78 ± 1.77	0.907

* Results are expressed as g/100 g; ** Istrian-type sausage with additional ingredients; Results are expressed as the mean value ± standard deviation; ^a,b^ values within a row with no common superscript differ significantly (*p* < 0.05); *p*-value refers to results of analyzed parameter per row among five sausage types.

**Table 5 foods-09-00990-t005:** Fatty acids and fat quality indices descriptive of Croatian traditional dry-fermented sausages.

Parameters	Istrian Domestic Sausage (*n* = 36)	Kosnica (*n* = 21)	Istrian-Type Sausage * (*n* = 33)	Slavonian Domestic Sausage (*n* = 27)	Kulenova Seka (*n* = 24)	*p*-Value
**Fatty acids**						
C10:0	0.10 ± 0.01 ^ab^	0.10 ± 0.01 ^a^	0.10 ± 0.01 ^ab^	0.10 ± 0.01 ^ab^	0.08 ± 0.01 ^b^	0.021
C12:0	0.10 ± 0.03	0.10 ± 0.03	0.11 ± 0.04	0.08 ± 0.00	0.08 ± 0.01	0.060
C14:0	1.44 ± 0.10 ^ab^	1.41 ± 0.11 ^ab^	1.58 ± 0.26 ^a^	1.34 ± 0.05 ^b^	1.29 ± 0.10 ^b^	0.002
C15:0	0.06 ± 0.01 ^b^	0.06 ± 0.02 ^ab^	0.12 ± 0.09 ^a^	0.05 ± 0.02 ^b^	0.05 ± 0.02 ^b^	0.007
C16:0	25.88 ± 0.67 ^a^	26.16 ± 0.71 ^a^	25.58 ± 1.16 ^ab^	24.99 ± 0.99 ^ab^	24.38 ± 1.40 ^b^	0.006
C16:1 n-7*t*	0.40 ± 0.06 ^a^	0.35 ± 0.05 ^ab^	0.40 ± 0.08 ^ab^	0.32 ± 0.05 ^b^	0.38 ± 0.04 ^ab^	0.008
C16:1n-7*c*	2.53 ± 0.32	2.47 ± 0.25	2.53 ± 0.48	2.37 ± 0.19	2.67 ± 0.50	0.606
C17:0	0.52 ± 0.07	0.53 ± 0.10	0.71 ± 0.35	0.50 ± 0.11	0.53 ± 0.12	0.077
C17:1	0.30 ± 0.06	0.31 ± 0.04	0.35 ± 0.13	0.28 ± 0.05	0.34 ± 0.09	0.272
C18:0	13.96 ± 1.06 ^ab^	14.27 ± 0.88 ^ab^	14.68 ± 1.38 ^a^	13.91 ± 1.01 ^ab^	12.60 ± 1.72 ^b^	0.016
C18:1n-9*t*	0.17 ± 0.05 ^ab^	0.17 ± 0.04 ^ab^	0.23 ± 0.08 ^a^	0.16 ± 0.03 ^b^	0.19 ± 0.05 ^ab^	0.033
C18:1n-9*c*	39.97 ± 2.04 ^ab^	41.28 ± 1.81 ^ab^	39.45 ± 2.41 ^b^	40.07 ± 1.73 ^ab^	42.93 ± 2.67 ^a^	0.012
C18:1n-7	2.93 ± 0.26	2.86 ± 0.22	2.70 ± 0.34	2.76 ± 0.24	3.01 ± 1.29	0.786
C18:2n-6*t*	0.13 ± 0.02	0.13 ± 0.01	0.16 ± 0.06	0.12 ± 0.01	0.14 ± 0.02	0.079
C18:2n-6*c*	9.22 ± 2.28	7.73 ± 2.04	8.90 ± 2.02	10.41 ± 1.90	8.73 ± 1.96	0.152
C18:3n-3	0.43 ± 0.16 ^ab^	0.32 ± 0.10 ^b^	0.59 ± 0.30 ^a^	0.56 ± 0.21 ^ab^	0.38 ± 0.20 ^ab^	0.043
C20:0	0.21 ± 0.02	0.21 ± 0.02	0.21 ± 0.02	0.21 ± 0.01	0.22 ± 0.04	0.871
C20:1n-9	0.83 ± 0.06 ^b^	0.83 ± 0.05 ^b^	0.81 ± 0.14 ^b^	0.82 ± 0.09 ^b^	1.01 ± 0.11 ^a^	0.001
C20:2n-6	0.41 ± 0.09 ^ab^	0.35 ± 0.07 ^b^	0.39 ± 0.08 ^ab^	0.47 ± 0.06 ^a^	0.45 ± 0.09 ^ab^	0.045
C20:4n-6	0.15 ± 0.03 ^b^	0.10 ± 0.01 ^c^	0.13 ± 0.05 ^bc^	0.22 ± 0.05 ^a^	0.20 ± 0.09 ^abc^	0.002
C20:3n-3	<LOD	<LOD	0.06 ± 0.08	0.10 ± 0.05	0.05 ± 0.07	0.215
C23:0	0.16 ± 0.05 ^a^	0.22 ± 0.05 ^a^	0.07 ± 0.07 ^ab^	0.05 ± 0.05 ^b^	0.13 ± 0.26 ^ab^	0.000
SFA	42.43 ± 1.49 ^ab^	43.06 ± 1.49 ^a^	43.17 ± 2.13 ^a^	41.21 ± 1.99 ^ab^	39.37 ± 3.33 ^b^	0.008
MUFA	47.13 ± 2.36 ^ab^	48.27 ± 2.02 ^ab^	46.51 ± 2.84 ^b^	46.78 ± 2.06 ^ab^	50.53 ± 4.06 ^a^	0.025
n-6	9.92 ± 2.36	8.31 ± 2.11	10.28 ± 2.32	11.27 ± 2.03	10.08 ± 2.32	0.121
n-3	0.43 ± 0.16 ^ab^	0.32 ± 0.10 ^b^	0.70 ± 0.37 ^ab^	0.73 ± 0.27 ^a^	0.50 ± 0.29 ^ab^	0.007
PUFA	10.34 ± 2.48	8.62 ± 2.19	9.58 ± 2.10	11.99 ± 2.25	9.58 ± 2.07	0.097
**Fat quality indices**						
Total TFA	0.70 ± 0.09 ^ab^	0.65 ± 0.09 ^ab^	0.79 ± 0.14 ^a^	0.59 ± 0.07 ^b^	0.72 ± 0.09 ^ab^	0.003
n-6/n-3	24.88 ± 6.58 ^a^	26.74 ± 4.21 ^a^	16.56 ± 7.26 ^b^	16.50 ± 3.82 ^b^	22.25 ± 6.68 ^ab^	0.001
PUFA/SFA	0.24 ± 0.06	0.20 ± 0.05	0.24 ± 0.06	0.29 ± 0.07	0.26 ± 0.06	0.061
IA	0.55 ± 0.03 ^a^	0.56 ± 0.03 ^a^	0.57 ± 0.05 ^a^	0.52 ± 0.04 ^ab^	0.49 ± 0.06 ^b^	0.001
H/H	1.82 ± 0.09 ^b^	1.79 ± 0.10 ^b^	1.81 ± 0.15 ^b^	1.95 ± 0.14 ^ab^	2.05 ± 0.20 ^a^	0.004
IT	1.39 ± 0.09 ^ab^	1.43 ± 0.10 ^a^	1.39 ± 0.11 ^ab^	1.29 ± 0.12 ^ab^	1.22 ± 0.17 ^b^	0.005

* Istrian-type sausage with additional ingredients; Results are expressed as the mean value in% of total fatty acids ± standard deviation; LOD—limit of detection = 0.05%; SFA—saturated fatty acids, MUFA—monounsaturated fatty acids, PUFA—polyunsaturated fatty acids; TFA—trans fatty acids; H/H—hypo-/hyper-cholesterolemic fatty acids ratio, AI—atherogenic index, TI—thrombogenic index; ^a–c^ values within a row with no common superscript differ significantly (*p* < 0.05); *p*-value refers to results of analyzed parameter per row among five sausage types.

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
