# Peer review of "Characterization of Traditional Croatian Household-Produced Dry-Fermented Sausages"

_foods, 2020, doi:10.3390/foods9080990_

Round 1

Reviewer 1 Report

foods-863292

Characterisation of Traditional Croatian Household-Produced Dry-Fermented Sausages

The authors present a nicely done study that characterized microbial, chemical, and sensory properties of 5 different types of sausage.  The methods are appropriate, the results sound, and their implications well discussed.  Apart from small word use /grammar items that can be addressed by a proof reader, I only have comments regarding how Tables present the results and statistical significance is indicated to the reader; and Figures 1 and 2 are difficult to identify which points are different from others.

Specifically,

Line 237, Table 2 and Line 245.  It is not clear at all what exactly the values in the table represent, nor what is significantly different between sausage results in this table.

If these are Number of sausage infected, out of how many?  Presenting a percentage (%) rather than a number would be good.  Or are these Number of isolates?  That still leave confusion.

Please revise table for clarity, and if statistical differences are to be indicated, add indicators within rows where appropriate.  Indicators should be rank order applied highest value to lowest, where values followed by the  same indicating letter are not different (P>0.05).

Line 276 & 279: Figures 1 and 2: I appreciate this sort of data plot to summarize all of the  variables rather than a very dense table, but the plot lines are too heavy or too great a point, so that differences between them cannot be distinguished by the reader.  Try hairline, 0.5, or 1.0 point lines to improve the readability of these two figures.

Line 370: Table 3.  A column of number (n) of each type of sausage should be added.

Also, the statistical indicators with in each column are confusing in regards to the footnotes. Please revise table for clarity, for statistical differences add indicators within columns that are in rank order applied highest value to lowest, where values followed by the same indicating letter are not different (P>0.05).

Line 383 and 449: Tables 4 and 5.  Consider adding the ‘n’ for each group of sausage below its title, then within each row add statistical indicators in rank order applied highest value to lowest, where values with in the same row followed by the same indicating letter are not different (P>0.05).  What does the P value refer too? Which comparison? All combined?  It is better to provide clear statistical indicators for reader. 

Author Response

Comments of the Referee 1:

Comment: Line 237, Table 2 and Line 245. It is not clear at all what exactly the values in the table represent, nor what is significantly different between sausage results in this table. If these are Number of sausage infected, out of how many? Presenting a percentage (%) rather than a number would be good. Or are these Number of isolates? That still leave confusion. Please revise table for clarity, and if statistical differences are to be indicated, add indicators within rows where appropriate. Indicators should be rank order applied highest value to lowest, where values followed by the same indicating letter are not different (P>0.05).

Answer: We agree with the Reviewer that the results displayed in Table 2 were not given clearly, so that the Table has been revised. As suggested by the Reviewer, in the revised Table the number of mould isolates is replaced by the percentage of the latter, displayed based both on mould and on product type. Given the inappropriateness of tagging the values yielded by statistical evaluation with indicators within the table row, data on statistical significance are now given below the Table based on the type of data processing (type of the sausage, mould genus and region of origin).

Comment: Line 276 & 279: Figures 1 and 2: I appreciate this sort of data plot to summarize all of the  variables rather than a very dense table, but the plot lines are too heavy or too great a point, so that differences between them cannot be distinguished by the reader.  Try hairline, 0.5, or 1.0 point lines to improve the readability of these two figures.

Answer: In the revised manuscript, Figures 1 and 2 have been plotted using 0.5-point lines, which, we believe, gave rise to distinctiveness, i.e. clarity of presentation of objective and subjective sensorial characteristics of sausages.

Comment: Line 370: Table 3. A column of number (n) of each type of sausage should be added. Also, the statistical indicators with in each column are confusing in regards to the footnotes. Please revise table for clarity, for statistical differences add indicators within columns that are in rank order applied highest value to lowest, where values followed by the same indicating letter are not different (P>0.05).

Answer: We concur with the Reviewer that the results given in Table 3 were not displayed in a comprehensive manner, so that the Table has been supplemented with a column containing the number of samples of each sausage type. On top of that, as recommended by the Reviewer, the manner of statistical significance tagging has been changed; in the revised manuscript, indicators within columns where values followed by the same indicating letter are not different.

Comment: Line 383 and 449: Tables 4 and 5. Consider adding the ‘n’ for each group of sausage below its title, then within each row add statistical indicators in rank order applied highest value to lowest, where values with in the same row followed by the same indicating letter are not different (P>0.05). What does the P value refer too? Which comparison? All combined?  It is better to provide clear statistical indicators for reader.

Answer: We agree with the Reviewer's opinion, so that Tables 4 and 5 are now supplemented with data on the number of samples of each type of sausage. On top of that, in each row statistical indicators are added in rank order from the highest value to the lowest, where values within the same row followed by the same indicating letter are not different. Beneath the Tables, explanation as to the pertinence and cross-reference of the displayed p-values has been given, as well.

The authors are indebted to our esteemed Reviewers for their most helpful suggestions and comments.

The manuscript has been thoroughly rechecked by a highly qualified linguist proficient in scientific writing, hopefully to the Reviewer’s satisfaction.

Reviewer 2 Report

The manuscript reports the results of a survey on the characterization of five types of dry fermented sausages traditionally produced in Croatia.

The study is a simple presentation of some physico-chemical properties as well as fatty acid composition, mycobiota identification and sensory evaluation of the dry fermented sausages gathered in 2018 and 2019 and produced by 47 family farms.

The scientific approach of this study is limited because it is a simple description of compositional traits. However, no data on the Croatian dry fermented sausages are available in literature.

The presentation of the Results and Discussion section is, in some part, too long and general. General sentences might be removed.

Some amendments:

Line 36 and 56: some international studies on traditional dry fermented sausages from Central and South Europe should be mentioned (e.g. Demeyer et al. (2000). Food Research International, 33, 171-180; 89.          Talon et al., (2012) Chapter 6. Microbial ecosystem of traditional dry fermented sausages in Mediterranean countries and Slovakia. In Williams G.S. Ed. Mediterranean ecosystems: dynamics, management and conservation. Nova Science Publishers Inc., Hauppauge, NY (ISBN 978-1-61209-146-4), pp. 115-128).

Line 158: how it is possible to measure color parameters by colorimeter in homogenized sample? For dry fermented sausages is recommended to measure the color parameters in different part of the whole slice.  

Author Response

Comments of the Referee 2:

Comment: Line 36 and 56: some international studies on traditional dry fermented sausages from Central and South Europe should be mentioned (e.g. Demeyer et al. (2000). Food Research International, 33, 171-180; 89.; Talon et al., (2012) Chapter 6. Microbial ecosystem of traditional dry fermented sausages in Mediterranean countries and Slovakia. In Williams G.S. Ed. Mediterranean ecosystems: dynamics, management and conservation. Nova Science Publishers Inc., Hauppauge, NY (ISBN 978-1-61209-146-4), pp. 115-128).

Answer: We are obliged to the Reviewer for the extremely useful references he/she drew our attention to. The suggested references have now been quoted in the revised body text (lines 36 and 56). Due to the inclusion of the references in question, their markings have been changed throughout the revised body text.

Comment: Line 158: how it is possible to measure color parameters by colorimeter in homogenized sample? For dry fermented sausages is recommended to measure the color parameters in different part of the whole slice.

Answer: We are obliged to the Reviewer for this comment. Colour determination made use of homogenised samples given that the results of our research have shown the closeness of average colour parameters of homogenised and whole-slice dry-fermented sausage samples. Of note, some authors whose studies also included dry-sausage colour measurements, have resorted to homogenization, too  (i.e. Ansorena, De Pefia, Astiasarhn, Bello (1997) Application of modem colour systems in investigation of colour changes in dry fermented sausages during production, Meat Science, 46, 313-318).

The authors are indebted to our esteemed Reviewers for their most helpful suggestions and comments.

The manuscript has been thoroughly rechecked by a highly qualified linguist proficient in scientific writing, hopefully to the Reviewer’s satisfaction.